# The effects of haloperidol on motor vigour and movement fusion during sequential reaching

Sebastian Sporn[1,2]*, Joseph M. Galea[1]

**1** School of Psychology and Centre for Human Brain Health, University of Birmingham, Birmingham, United Kingdom, **2** Department of Clinical and Movement Neuroscience, Queens Square Institute of Neurology, UCL, London, United Kingdom

* spornseb@gmail.com

**Data Availability Statement:** Analysis code is available on the Open Science Framework website, alongside the experimental datasets at: https://osf.io/62wcz/.

## Abstract

Reward is a powerful tool to enhance human motor behaviour with previous research showing that during a sequential reaching movement, a monetary incentive leads to increased speed of each movement (motor vigour effect), whilst reward-based performance feedback increases the speed of transition between movements (movement fusion effect). The neurotransmitter dopamine plays a central role in the processing of reward signals and has been implicated to modulate motor vigour and regulate movement fusion. However, in humans, it is unclear if the same dopaminergic mechanism underlies both processes. To address this, we used a complex sequential reaching task in which rewards were based on movement times (MT). Crucially, MTs could be reduced via: 1) enhanced speed of individual movements (motor vigour effect) and/or 2) enhanced speed of transition between movements (movement fusion effect). 95 participants were randomly assigned to a reward or no reward group and were given either 2.5mg of the dopamine antagonist haloperidol or a placebo (control group). An independent decision-making task performed prior to the main experiment suggested that haloperidol was active during the sequential reaching task (positive control). We did not find evidence that haloperidol affected the facilitatory effects of reward on movement fusion. However, we found that haloperidol negated the reward-based effects on motor vigour. Therefore, our results suggest that a D2-antagonist differentially influences reward-based effects on movement vigour and movement fusion, indicating that the dopaminergic mechanisms underlying these two processes may be distinct.

## Introduction

Ubiquitous in daily life, but often impaired in clinical populations [1–3], the execution of motor sequences underlies a vast range of behaviours such as drinking a cup of coffee to driving a car. Reward has been demonstrated to promote motor sequence learning with research showing that enhanced motor sequence learning allows for a faster and smoother, yet similarly accurate execution of the motor sequence [4–9]. Therefore, reward may represent a powerful tool to promote motor sequence learning in rehabilitation settings.

**Funding:** This work supported by the European Research Council. The funders had no role in study design, data collection and analysis, decision to publish, or preparation of the manuscript.

**Competing interests:** The authors have declared that no competing interests exist.

Importantly, recent work found that different reward types lead to dissociable motor improvements [10, 11]. Specifically, while providing money independent of performance feedback enhanced movement speed (motor vigour effect), providing feedback/points independent of money improved movement fusion (movement quality effect) [10]. Movement fusion represents a learning-dependent optimisation process during which individual motor elements of a sequence are blended into a combined singular action [10, 12–14]. Importantly, fusion has been demonstrated to lead to improvement in movement quality through enhanced smoothness/decreased jerk [10, 12–14]. Thus, reward appears to improve two core aspects of motor control during the production of sequential movements, leading to improvements in speed and efficiency [15]. Yet, it is unclear if these two reward-based processes have dissociable neural mechanisms, which is of vital importance for reward to be employed in a strategic and targeted manner in clinical settings such as rehabilitation.

It has long been established that the neurotransmitter dopamine (DA) plays a central role in the processing of reward signals [16–18] and has been suggested to modulate reaching speed (motor vigour) [19, 20]. Experimental evidence comes from both animal [21, 22] and human [23–30] work. Specifically, L-DOPA has been shown to increase motor vigour through increased willingness to exert motor effort [23–26], with corroborating evidence coming from Parkinson's patients (PD) [27–30]. Conversely, research using haloperidol demonstrated a reduction in motor vigour during reward-effort decision-making [26]. Haloperidol is a D2-receptor antagonist that shows a limited affinity to D1 receptors and has superior in vivo D2 binding. It blocks DA D2 binding in the basal ganglia (BG) but not in the prefrontal cortex and as such can be considered to selectively modulate DA levels within the BG pathway [31]. Additionally, DA has been implicated to underlie movement fusion (chunking) with evidence coming from rodent [32–34], monkey [1, 35] and human [1, 3, 36–41] work. Particularly, PD patients OFF medication showed impaired motor sequence learning [1, 37–39, 41] and reduced movement anticipation [3, 40], which is a prerequisite of movement fusion [42]. Additionally, research using tiapride, a D2 antagonist, found impaired motor sequence learning during a button-press task [36]. Therefore, DA appears to play a crucial role in two core aspects of motor sequence learning which have also been shown to be reward sensitive. Importantly, while changes in tonic DA are believed to underlie the modulation of motor vigour, phasic DA has been suggested to underpin movement fusion (chunking) [20, 43]. Within this context, it may seem plausible that the reward-based processes driven by monetary incentives and performance-based feedback are indeed based on dissociable dopaminergic mechanisms. In contrast, research using D2 antagonists suggests that motor vigour and movement fusion may in fact rely on the same DA receptors within the corticostriatal circuitry [26, 36, 44]. However, to date neither possibility has been experimentally tested in humans.

To address this, we used a complex sequential reaching task in which participants were asked to execute a continuous sequence of eight reaching movements [10]. Participants received a combination of two rewards (i.e., monetary incentive and performance-based feedback) that have been shown to distinctively enhance motor sequence learning by increasing vigour and fusion. To investigate whether these reward-based processes rely on the same or dissociable dopaminergic mechanisms, participants were randomly assigned to a reward or no reward group and were given either 2.5mg of the D2 antagonist haloperidol or a placebo.

## Methods

### Participants

From February–October 2019, 95 participants (42 males and 53 females; age range 18–42) were recruited to participate in three sessions, which had been approved by the local research

ethics committee of the University of Birmingham. All participants were novices to the task paradigm and were free of motor, visual and cognitive impairment. Participants were pre-screened and were only invited to the medical exam if they met the following criteria: 1) naïve to the task paradigm; 2) 18–45 years old; 3) no self-reported history of medical disorders; 4) normal or corrected-to-normal vision; 5) no drug allergies; 6) currently taking no medication that interfere with the absorption of haloperidol. Suitable participants were then evaluated by a medical doctor, who reviewed their medical history, evaluated an electrocardiogram taken at rest and took a blood pressure reading. Participants who received medical approval were then scheduled for all experimental sessions. Most participants were self-reportedly right-handed (N = 88) while the rest was left-handed (N = 7) and gave written informed consent prior to the start of the experiment. Participants were remunerated with money (£18/hour) and were able to earn additional money during the task depending on their performance. Before the start of the experiment, participants were counterbalanced to one of the available groups.

## Experimental protocol, randomisation and blinding protocol

In this study, we sought to investigate whether DA modulates the reward-based improvement of movement vigour and/or movement fusion. To this end, participants were randomly allocated to one of four groups: haloperidol with reward-based feedback (Halo-R, N = 25), haloperidol without reward (Halo-NR, N = 24), placebo with reward-based feedback (Ctrl-R, N = 23) and placebo without reward (Ctrl-NR, N = 23) after the medical assessment (Fig 1C). Due to the required testing environment this study was single-blind, and both the medical doctor and examiner were aware of the drug group allocation (haloperidol vs placebo). However, to reduce bias all participants were told that they would receive either a placebo tablet or the active drug (haloperidol). Similarly, all participants had to complete a health check on the day of drug/placebo intake (*Day1*) and were checked by the medical doctor in intervals of 1h throughout Day1. Additionally, all task instructions were displayed on screen instead of communicated verbally to further reduce bias. The administration of both the active drug and the placebo tablet was performed by the medical doctor.

On the day of drug/placebo intake (*Day1*), participants either received 2.5mg of haloperidol (2 x 0.5mg and 1 x 1.5mg tablet) or three lactose tablets of the same white colouring. In each case, participants were handed an envelope containing either the active drug or placebo and were asked to close their eyes during intake. Haloperidol is a D2-receptor antagonist that shows a limited affinity to D1 receptors and has superior in vivo D2 binding. In addition, it blocks DA D2 binding in the basal ganglia (BG) but not in the prefrontal cortex and as such can be considered to selectively modulate DA levels within the BG pathway [31]. To coincide with the peak plasma concentration, participants were asked to wait in the lab for 120min before engaging in the motor task [31]. The chosen dose of 2.5mg of haloperidol and the waiting time of 2h were similar to previous studies that were able to observe drug-related behavioural and neurophysiological effects of haloperidol [31, 45]. After the waiting period, participants were asked to complete the motor task (see Task Design and Task Protocol for details) and upon completion were yet again checked by the medical doctor. On *Day1* participants were additionally asked to complete a questionnaire to report their perceived levels of fatigue and attention. We added this self-report because while haloperidol is a known D2 antagonist it may also act as sedative which could influence results. 42 participants (Halo = 21; Control = 21) completed the questionnaire (which was added while the trial was already ongoing) and answered the following questions: 'Please rate yourself with regards to: 1) Attention and 2) Fatigue'. The scale ranged from 1–8 with lower numbers indicating lower levels of attention and fatigue. Participants also completed a Probabilistic Go/No-Go task which was

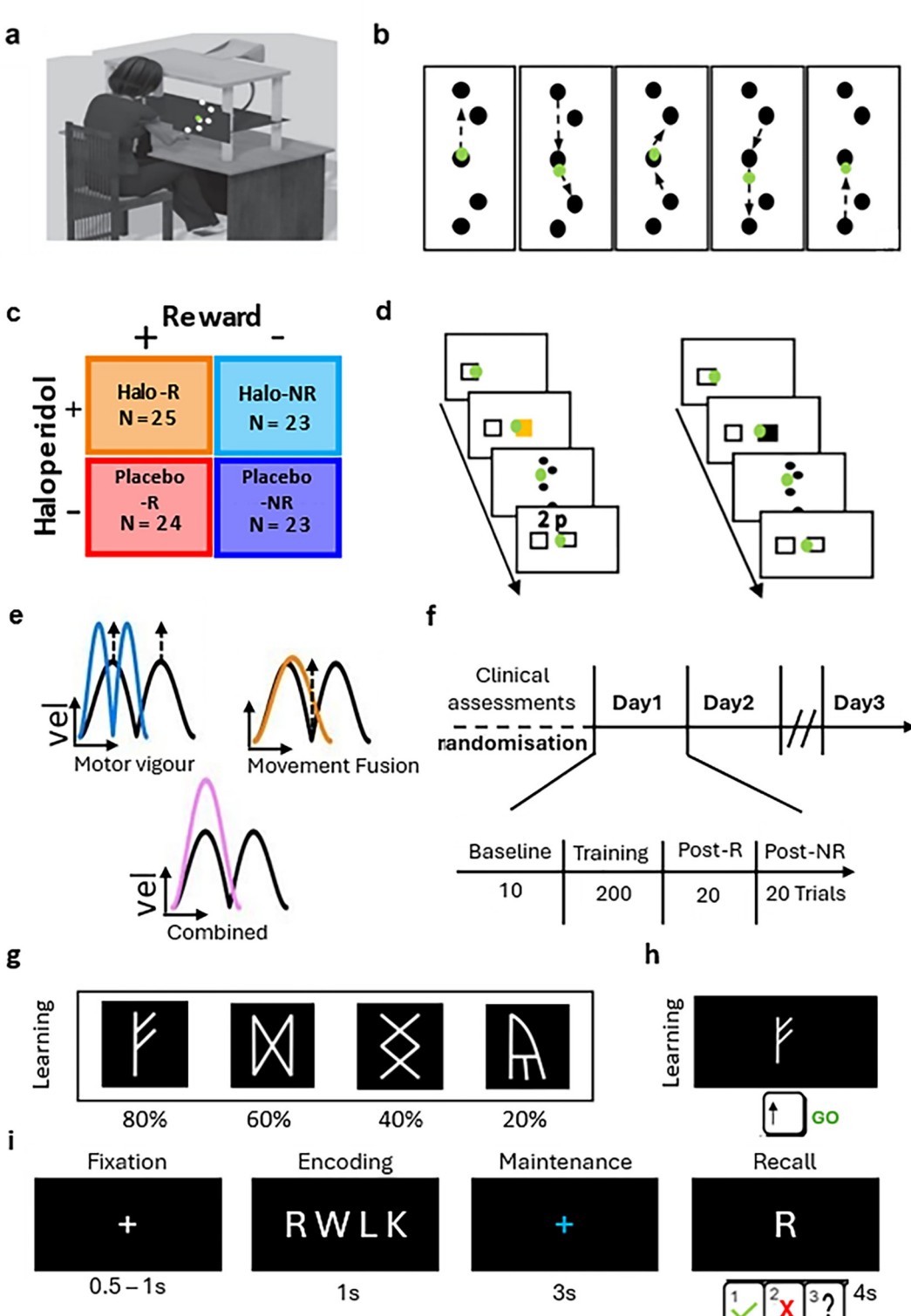

**Fig 1. Experimental setup. a)** Participants wore a motion-tracking device on the index finger and the unseen reaching movements were performed across a table whilst a green cursor matching the position of index finger was viewable on a screen. **b)** 8 movement sequential reaching task. Participants started from the centre target. **c)** Groups. Participants either received monetary incentive with accurate performance-based feedback (*Rew*) or not (*NoRew*). Similarly, participants were either received 2.5mg of haloperidol (*Halo*) or a placebo (*Control*). **d)** Trial design. Reward trials (left) were cued using a visual

stimulus (yellow start box) prior to the start of the trial and participant received performance-based feedback depending on how fast they completed the trial. In contrast, no reward trials were not cued, and no feedback was given (right). Instead, participants were asked to complete each trial as fast and accurately as possible. **e)** Strategies. To reduce MTs to earn more money participants could pursue two independent strategies: 1) Increase peak velocities (top left) and 2) Fuse consecutive movements leading to increases in minimum velocities (top right). Note these strategies are not mutually exclusive and can be combined to further reduce MTs (bottom). **f)** Study design. Prior to the start of the experiment, participants were trained on the reaching sequence and were then asked to perform 10 baseline trials. Randomly allocated to one of four groups, participants completed 200 trials during *Training* and an additional 20 trials in each *Post Assessment*; one with reward (*post-Rew*) and one without (*post-NoRew*). This design was repeated on *Day2* (without drug intake) and on *Day7* participants engaged in a *post-Rew* and *post-NoRew* (25 trials). **g)** Stimuli (Celtic runes) used in the *Learning* phase of the Probabilistic Go/No-Go Task, which minimised explicit verbal encoding, were presented in random order and were associated with reward probabilities. Participant were instructed to maximise points. **h)** By pressing the up-arrow key, participant collected the presented rune (Go) and received feedback on whether this led to a gain or loss in points. No performance-based feedback was given, after withholding a response (No-Go) by not pressing a key. **i)** Illustration of Verbal Working Memory (VWM) task. Each trial consisted of four phases (Fixation, Encoding, Maintenance, and Recall) with the time allocated to each displayed below. Participants had to make a decision during recall whether the presented consonant was and press either '1' to indicate that it was part of the array seen during encoding, '2' was not part of the array, and '3' not sure about the correct answer.

based on previous work which found that haloperidol impairs learning from negative feedback, while it enhanced learning from positive feedback compared to placebo (positive control) [31]. Participants were scheduled to return to the lab 24h later (*Day2*) to complete the same motor task; this time without any drug/placebo manipulation. However, participants received the same feedback as during *Day1* (i.e., with or without reward-based feedback during Training). Lastly, participants engaged in a short version of the motor task one week after the initial session (*Day7*) and were subsequently debriefed.

## Experimental apparatus

The experiment was performed using a Polhemus 3SPACE Fastrak tracking device (Colchester, Vermont U.S.A; with a sampling rate of 110Hz). Participants were seated in front of the experimental apparatus which included a table, a horizontally placed mirror 25cm above the table and a screen (Fig 1A). A low-latency Apple Cinema screen was placed 25cm above the mirror and displayed the workspace and participants' hand position (represented by a green cursor–diameter 1cm). On the table, participants were asked to perform 2-D reaching movements. Looking into the mirror, they were able to see the representation of their hand position reflected from the screen above. This setup effectively blocked their hand from sight. The experiment was run using MATLAB (The Mathworks, Natwick, MA), with Psychophysics Toolbox 3.

## Design of motor task

Participants were asked to hit a series of targets displayed on the screen (Fig 1A and 1B). Four circular (1cm diameter) targets were arranged around a centre target ('via target'). Starting in the via target, participants had to perform eight continuous reaching movements to complete a trial. Target 1 and 4 were displaced by 10cm on the *y*-axis, whereas Target 2 and 3 were 5cm away from the via target with an angle of 126 degrees between them (Fig 1B). To start each trial, participants had to pass their cursor though the preparation box (2x2cm) on the left side of the workspace, which triggered the appearance of the start box (2x2cm) in the centre of the screen. After moving the cursor into the start box, participants had to wait for 1.5s for the targets to appear. This ensured that participants were stationary before reaching for the first target. Target appearance served as the go-signal and the start box turned into the via target (circle). Upon reaching the last target (via target), all targets disappeared, and participants had to wait for 1.5s before being allowed to exit the start box to reach for the preparation box to

initiate a new trial. Participants had to repeat a trial if they missed a target or performed the reaching order incorrectly. Similarly, exiting the start box too early either at the beginning or at the end of each trial resulted in a missed trial.

## Reward structure and feedback

Participants in the *Rew* groups were informed that faster MTs would earn them more money. Reward trials were cued using a visual stimulus prior to the start of the trial (Fig 1D left). Once participants moved into the preparation box, the start box appeared in yellow (visual stimulus). In contrast, participants that were in a *NoRew* groups were told to 'move as fast and accurately as possible' (Fig 1D right). Performance feedback was provided after completing a trial while participants moved from the start box to the preparation box to initiate a new trial. Feedback was displayed on the top of the screen (i.e., '2p out of 5p'). We used a closed-loop design to calculate the feedback in each trial. To calculate this, we included the MT values of the last 20 trials and organised them from fastest to slowest to determine the rank of the current trial within the given array. A rank in the top three ($< = 90\%$) returned a value of 5p, ranks $> = 80\%$ and $<90\%$ were valued at 4p; ranks $> = 60\%$ and $<80\%$ were awarded 3p; ranks $> = 40\%$ and $< 60\%$ earned 2p while 1p was awarded for ranks $> = 20\%$ and $< 40\%$. A rank in the bottom three ($<20\%$) returned a value of 0p. When participants started a new experimental block, performance in the first trial was compared to the last 20 trials of the previously completed block.

## Task protocol

The main experiment included four experimental parts: *Baseline*, *Training*, a *Post Assessment* with reward and one without (Fig 1F). Participants completed this protocol on *Day1* and *Day2*, while only completing the two post assessments on *Day7*. Additionally, a learning block was scheduled prior to the start of the main experiment on *Day1*. Furthermore, a secondary task was included in this task design, which asked participants to press a force sensor with the index finger of their non-dominant hand. Participants were told to apply pressure in response to an audio signal that changed in amplitude, with higher amplitudes requiring increased force and vice versa. However, the analysis of these secondary-task trials which were scheduled on every $10^{th}$ trial during training and every $5^{th}$ trial during the remaining parts are excluded from the analysis presented here.

**Familiarisation.** We included a learning phase prior to the start of the experiment on *Day1* so that participants could memorise the reaching sequence (Fig 1B). After completing 20 learning trials, participants moved on to the main experiment.

**Baseline.** Participants in both groups completed 10 baseline trials, which were used to assess whether there were any pre-training differences between groups. Both groups were instructed to 'move as fast and accurately as possible', while no performance-based feedback was given at the end of each trial.

**Training.** Participants in the *Rew* groups were informed that during this part they would be able to earn money depending on how fast they complete each trial (200 reward trials). In contrast, participants in the *NoRew* group engaged in 200 no reward trials and were again instructed to move as fast and as accurately as possible.

**Post assessments.** Participants from both groups were asked to complete two post assessments (20 trials each on *Day1* and *Day2* and 25 trials on *Day7*); one with reward trials (*post-Rew*) and one with no reward trials (*post-NoRew*). The order was counter-balanced across participants.

Consequently, participants completed three experimental sessions (*Day1*, *Day2* and *Day7*) as well as a medical assessment prior to the start of the study (Fig 1F).

**Dual task.** Additionally, a secondary task was included in this task design, which asked participants to press a force sensor with the index finger of their non-dominant hand. Participants were told to apply pressure in response to an audio signal that changed in amplitude, with higher amplitudes requiring increased force and vice versa. However, the analysis of these secondary-task trials which were scheduled on every 10th trial during training and every 5th trial during the remaining parts are excluded from the analysis presented here.

## Design of positive control

**Probabilistic Go/NoGo task.** The positive control was based on previous work showing that haloperidol impairs learning from negative feedback, while it enhanced learning from positive feedback compared to placebo [31]. Here, we simplified this Probabilistic Go/No-Go task using four stimuli (Celtic runes) which were associated with distinct reward probabilities (Fig 1G) and were presented in random order. Participants were instructed to either press the up-arrow key (Go) or withhold their response (No-Go, Fig 1H). They were told that some runes yield a point if selected, whereas others led to a loss of a point, and they were instructed to maximise points. Performance-based feedback was only provided after Go responses ('You won a point!', or 'You lost a point'), while 'No action recorded' was displayed after a No-Go response. Over the course of the *Learning* phase (40 trials; each rune was presented 10 times) participants should learn that two runes were associated with rewards, whereas the others led to losses (for more detail on the Probabilistic Go/No-Go task see [31]).

**Verbal memory task.** The effects of haloperidol on task performance in the Probabilistic Go/NoGo task were most apparent in participants with low working memory scores [31]. We, therefore, included a Verbal Memory Task (VMT) that was previously described in [46]. Specifically, the task consisted out of three parts (Fig 1I): 1) the encoding period (1s), during which participants were presented with a array of consonants, 2) maintenance period (3s), during which a blue fixation cross was displayed on the screen and 3) recall period, during which a single consonant was presented for a maximum of 4s. Participants had to decide if the presented consonant was part of the previously seen array (i.e., "Was this letter included in the previous array?"). Participants responded by pressing one of three keys on a keyboard with their dominant hand. '1' indicated that the consonant presented in the recall period was a part of the array (*match*); '2' indicated a *nonmatch*; and '3' indicated that the participant was unsure about the correct answer. Difficulty in this task was determined by the length of the array, ranging from 5 to 9. The VWM task consisted of 60 trials, and before the start of the task, participants performed 10 practice trials to familiarise themselves with the task and instructions.

**Data analysis.** Analysis code is available on the Open Science Framework website, alongside the experimental datasets at: https://osf.io/62wcz/. The analyses were performed in Matlab (Mathworks, Natick, MA) and JASP.

## Positive control

**Working memory performance (WM Perf).** WM Perf was used to identify participants with low and high WM (based on a median split) to be included in the subsequent analysis of the Probabilistic Go/NoGo task. To this end, participants whose average percentage of correct responses across the three highest levels of difficulty (7–9 letter long arrays) was below the median across all participants, were considered as low WM participants (N = 44; Halo_-Low = 24 and Ctrl_Low = 20). Conversely, participants whose average percentage of correct

 

responses was above the median, were considered as high WM participants (N = 48; Halo_-High = 20 and Ctrl_High = 23).

**Response Accuracy (RA).** Response accuracy was used to determine whether participants on haloperidol learned more from positive or negative feedback than controls. This positive control was included to ascertain if the drug manipulation worked. To this end, Go and No-Go responses during Learning were assessed in trials that were either strongly associated with rewards or losses. Therefore, RA (in %) reflects how often participants triggered a Go response in trials that were more likely to yield rewards (positive feedback learning) and a No-Go response in trials that were more likely to led to losses (negative feedback learning). Importantly, similarly to previous work [31], this analysis was done separately for low and high WM participants.

## Motor experiment

**Movement time (MT).** MT was measured as the time between exiting the start box and reaching the last target. This includes reaction time, which describes the time between target appearance and when the participants' start position exceeded 2cm.

**Maximum velocity ($v_{max}$).** Through the derivative of positional data $(x, y)$, we obtained velocity profiles for each trial which were smoothed using a gaussian smoothing kernel ($\sigma = 2$). The velocity profile was then divided into segments representing movements to each individual target (8 segments). Specifically, a segment was defined as when the cursor left the target until it left the next. Note here that the final segment was defined as when the cursor left the previous target until it reached the final target. We measured the maximum velocity ($v_{max}$) of each segment by finding the maximum velocity:

$$v_{max} \triangleq max_{\epsilon[t_1 \ t_2]} v(t) \qquad \text{Eq1}$$

Where $v(t)$ is the velocity of segment $t$, and $t_1$ and $t_2$ represent the start and end of segment $t$ respectively. The individual maximum velocities were then averaged for each trial.

**Fusion index (FI).** Fusion describes the blending together of individual motor elements into a singular smooth action. This is represented in the velocity profile by the velocity trough between the two movements gradually increasing [12–14]. To measure fusion, we compared the mean maximum velocities of two sequential reaches with the minimum velocity around the via point. The smaller the difference between these values, the greater coarticulation had occurred between the two movements [47]. We calculated movement fusion by:

$$Fusion \ Index \triangleq 1 - \frac{\left(\frac{v_{max}1 + v_{max}2}{2}\right) - v_{min}}{\left(\frac{v_{max}1 + v_{max}2}{2}\right)} \qquad \text{Eq2}$$

with $v_{max}^1$ and $v_{max}^2$ representing the velocity peak value of two reaching movements, respectively, and $v_{min}$ representing the minimum value between these two points. We normalised the obtained difference, ranging from 0 to 1, with 1 indicating a fully fused movement. Given that in this task seven transitions had to be completed, we calculated the fusion index separately for each transition, which led to the maximum FI value being 7 in each trial.

**Data exclusion.** Assessing Baseline performance, we found that 3 participants exhibited >8.5s to complete a trial. We decided to remove these participants from further analysis after conducting an outlier analysis and comparing their mean (10.03s ± 1.0) to the rest of the participants (5.37s ± 1.1). In total, 2 participants from the Halo-NR group and 1 further participant from the Ctrl-NR group were removed. This changed the group sizes with Halo-R

 

(N = 25), Halo-NR (N = 21), Ctrl -R (N = 24) and Ctrl -NR (N = 22). Furthermore, trials that took longer than >10s were excluded from further analysis (N = 11). Finally, all dual task trials were removed from the analysis. Dual task trials were scheduled on every 10th trial during training and every 5th trial during the remaining parts which meant that for each participant 2 trials were removed from Baseline, 20 from Training and 4 and 5 during the post assessments on Day1/2 and Day7 respectively.

## Analysis plan

### Primary analyses.

1) Did the drug manipulation work?

To assess if the D2 antagonist haloperidol was active during the main experiment, we performed a positive control prior to its start. We used the same analysis pipeline as in [31]. Specifically, response accuracy (RA) was independently calculated for participants in the *Halo* and *Control* groups for both *Feedback Conditions*: positive feedback learning and negative feedback learning. To assess if the D2 antagonist haloperidol was active during the main experiment we used a mixed-model ANOVA with *Drug* (haloperidol vs placebo) as between factors and *Feedback Condition* (positive vs negative) as a within factor and RA as the dependent variable. Wilcoxon tests were employed when a significant interaction was reported.

2) Did haloperidol affect the reward-based effects on motor vigour?

In the main experiment, participants in the *Rew* groups were asked to complete a 'trial as fast and accurate as possible'. *Rew* participants received a combination of a monetary incentive and performance-based feedback. Importantly, participants could pursue two strategies to reduce MTs in this task: 1) increases in peak velocities of the individual movements (vigour effect) and 2) reductions in dwell-times when transitioning between movements via increases in minimum velocities (fusion effect; note here that these strategies are not mutually exclusive and can be combined to further reduce MTs—Fig 1E).

To investigate if haloperidol modulated the reward-based effects on $vel_{max}$, we used linear mixed-models (LMM), which were performed separately for *Training* on *Day1* (*Training1*) and *Day2* (*Training2*) using the following formula:

Dependent Variable ~ 1 + Reward*Drug*WM*Trial Number + (Trial|Participant)

LMMs were conducted in JASP and included four fixed effects: 1. *Reward* (1 = Reward; 2 = No Reward), 2. *Drug* (1 = haloperidol; 2 = placebo), 3. *Working Memory (WM;* 1 = high; 2 = low*)* and 4. *Trial Number* (1:180). By default, all the fixed effects and their interactions are included in the model. Therefore, the model included all possible interaction terms between the three main effects (i.e., *Reward*Drug, Reward*WM, Reward*Trial Number, Drug*WM, Drug*Trial Number, WM*Trial Number, Reward*Drug*WM, Reward*Drug*Trial Number, Reward*WM*Trial Number, Drug*WM*Trial Number Reward*Drug*WM*Trial Number*). Individual differences were accounted for by including *Participant* (1:92) as a random effect. By default, all the random effects corresponding to the fixed effects are included which corresponds to the "maximal random effects structure justified by the design" [48]. Therefore, the model included random slopes for all fixed and random effects combinations. However, all random slopes involving *Reward*, *Drug* and *WM* were removed from analysis because they do not vary within a participant (i.e., they refer to a group). In contrast, *Trial Number* was included as a random slope by participant because we expect that participants may show variable changes across trials; essentially reflecting differential improvements over the course of Training. Including *Trial Number* improved the model as seen in a lower AIC and BIC.

Removing *Trial Number* as a random slope by participant increased the BIC and we therefore used this model for all LMMs presented in the paper. Additionally, the model terms were tested with the Satterthwaite method and a restricted maximum likelihood method was used to fit the model.

Significant interactions that included *Trial Number* were further analysed by calculating the marginal effects of the fixed effects on the slope of the dependent variable (across trials). The beta coefficients (β) of the slopes and their corresponding 95% confidence intervals (CI) were used to perform pairwise z-tests to assess whether slopes were significantly different. Multiple comparisons were corrected using Bonferroni correction. In contrast, significant interactions that did not include *Trial Number* were further assessed using a between-groups ANOVA using the median across all trials for each participant as input. Furthermore, to determine during which trial window groups were significantly different, we used a row-shuffle non-parametric permutation test. We obtained the t-statistic of each permutation and subsequently determined whether the real t-statistic is statistically different from the permuted ones (p < 0.05). This allowed us to investigate during which trials groups were exhibiting significantly different performance.

3) Did haloperidol affect the reward-based effects on movement fusion?

Changes in movement fusion were operationalised as changes in the fusion index (FI) in a trial. To investigate if haloperidol modulated the reward-based effects on FI, we employed the same analysis pipeline as outlined in 2) with FI as the dependent variable.

4) Did haloperidol affect overall movement times?

To investigate if haloperidol modulated the reward-based effects on overall movement times, we employed the same analysis pipeline as outlined in 2) with MT as the dependent variable.

**Secondary analyses.**

5) Did haloperidol affect reward-based effects on any outcome measure when reward availability changes quickly (post assessments)?

The design of the main experiment was the same as in [10] and included *Post Assessments* (*post-Rew* and *post-NoRew*) which were scheduled after *Training*. Therefore, participants completed assessments on *Day1* and *Day2* and additionally on *Day7*. Here, we aimed to assess if haloperidol has an effect even when reward availability changes quickly (20 trials only). Mixed-model ANOVAs were used to assess statistical significance during the post assessments, with *Reward* (reward vs no reward), *Drug* (haloperidol vs placebo) and *WM* (high vs low) as between factors and *Timepoint* (*post-Rew* vs *post-NoRew*) as a within factor. We used one-sample Kolmogorov–Smirnov tests to test our data for normality and found that all measures were not normally distributed. Median values were therefore used as input in all mixed-model ANOVAs (similar to [10, 49]). Wilcoxon tests were employed when a significant interaction and/or main effects were reported. The results were corrected for multiple comparisons with false discovery rate Effect sizes were estimated using Cohen's d.

## Results

1) Results are in line with previous work on the effects of haloperidol on learning from feedback.

To assess if the D2 antagonist haloperidol was active during the main experiment, we performed a positive control prior to its start. Specifically, in line with previous work, we

conducted separate analysis for low and high WM participants using a median split and only included the low WM participants in the analysis [31]. Our results are in line previous work showing that haloperidol leads to enhanced learning from positive feedback, while it impairs learning from negative feedback compared to placebo in low WM participants (Fig 2A). Specifically, results from the mixed-effects ANOVA revealed a significant interaction between *Drug x Feedback Condition* during *Learning* (F = 5.94, p = 0.01911, $\eta^2$ = 0.12, Fig 2A) for low WM participants. We also found a main effect for *Feedback Condition* (F = 9.26, p = 0.0040, $\eta^2$ = 0.18), but not for *Drug* (F = 0.11, p = 0.7430, $\eta^2 < 0.01$). This suggests that haloperidol enhanced learning from positive feedback (Learn+), whilst impaired learning from negative feedback (Learn-), which is in line with previous work. Post-hoc analysis compared the difference ($\Delta$) between Learn+ and Learn- across groups. We found a significant difference between groups (Z = 2.36, p = 0.0183, $\eta^2$ = 0.74), which highlights that haloperidol was active during the experiment. Complementary results come from the high WM participants. Aligning with previous work [31], we did not find a significant interaction between *Drug x Feedback Condition* (F = 0.61, p = 0.4396, $\eta^2$ = 0.02, Fig 2B). Additionally, no main effects for *Drug* (F = 0.86, p = 0.3584, $\eta^2$ = 0.03) or *Feedback Condition* (F = 1.53, p = 0.2240, $\eta^2$ = 0.04) were found. Overall, this suggests that haloperidol was active during the main experiment.

Importantly, haloperidol is a known D2 antagonist and, therefore, may also act as sedative which could influence results. Consequently, we included a self-report asking participants (N = 42) to rate their perceived levels of fatigue and attention at the end of *Day1* (drug intake). Analysis of the self-reported data did not reveal any significant group difference for both fatigue (Wilcoxon test: Z = 0.69, p = 0.4935, $\eta^2$ = 0.11) and attention levels (Z = -1.05, p = 0.2955, $\eta^2$ = 0.16). These results suggest that any changes in performance due to haloperidol were not likely due to global changes in fatigue or attention.

2) Haloperidol negates the reward-based effects on motor vigour only on *Day1*.

We found no significant differences at Baseline (ANOVA; group: F = 5.79, p = 0.1223). We then conducted a separate LMM for *Day1* and *Day2* with $vel_{max}$ as the dependent variable. We found a significant main effect for both *Reward* ($t_{(84)}$ = 40.39, p < 0.001) and *Trial Number* ($t_{(84)}$ = 10.03, p = 0.002, S1 Table in S1 Appendix; Fig 2C) on *Day1*. Importantly, a significant three-way interaction between *Reward*, *Trial Number*, and *Drug* was found ($t_{(84)}$ = 4.08, p = 0.047). Estimating marginal effects, we found that Ctrl-R significantly increased $vel_{max}$ over the course of Training ($\beta$ = 2.4, CI [1.0 3.7], p < 0.001). In contrast, $vel_{max}$ decreased for Halo-R albeit not significantly ($\beta$ = -0.46, CI [-1.76 0.85], p = 0.49). The slopes for *NoRew* groups were positive but not significant (Halo-NR: $\beta$ = 1.28, CI [-0.15 2.7], p = 0.079; Ctrl-NR: $\beta$ = 1.27, CI [-0.14 2.7], p = 0.077). Running pairwise z-test comparisons we found the slopes to be significantly different between Ctrl-R and Halo-R (z = -2.90; p = 0.023) but did not find any other comparison to be significant. This suggests that haloperidol led to a selective decrease in reward-based motor vigour. We further investigated this interaction by running separate permutation analyses for the *Rew* (Fig 2D) and *NoRew* groups (Fig 2E). We found that the Halo-R group exhibited significantly lower $vel_{max}$ towards the end of *Training* on *Day1* when compared to Ctrl-R (Trial 170–180).

Crucially, we did not observe any differences for maximum velocity in the *NoRew* groups (Fig 2E), which indicates that haloperidol had a selective effect on reward-based motor vigour. Additionally, we found a significant interaction between *Trial Number* and *Drug* ($t_{(84)}$ = 4.00, p = 0.049). Estimating marginal effects, we found that the *Ctrl* groups significantly increased $vel_{max}$ over the course of Training ($\beta$ = 1.8, CI [0.8 2.8], p < 0.001), while the Halo groups did not ($\beta$ = 0.41, CI [-0.6 1.4], p = 0.41). Results from the pairwise z-test comparison showed that the slopes were significantly different (z = -1.99; p = 0.047). This suggests that haloperidol led

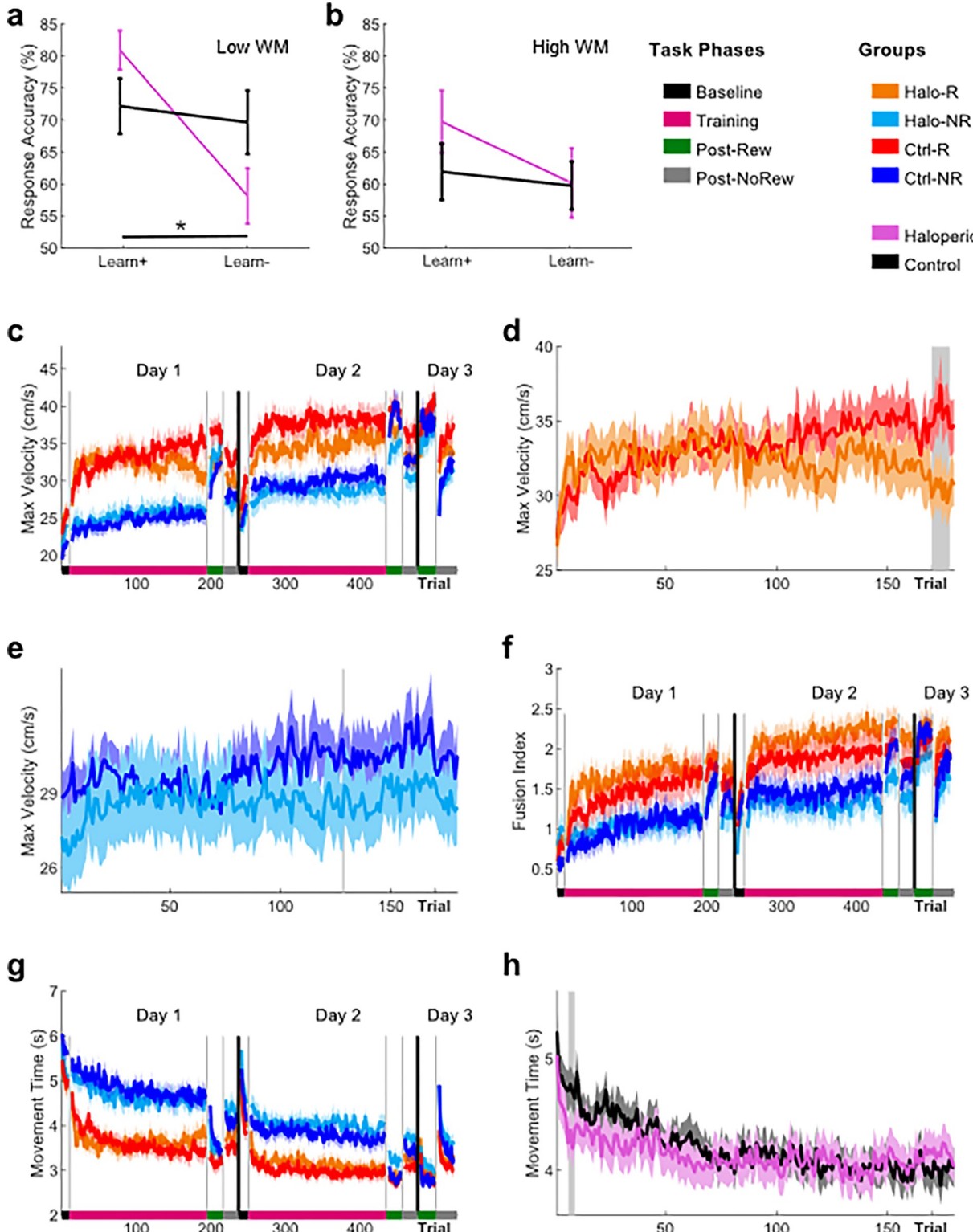

**Fig 2. Haloperidol only leads to a subtle impairment in peak velocities on Day1. a-b)** Mean Response Accuracy (%) in the Probabilistic Go/NoGoTrial task during **a)** Learning for positive (Learn+) and negative (Learn-) feedback and **b)** during Testing for both positive (Learn+) and negative (Learn-) feedback. Halo participants are in pink, while Control participants are in black. **c)** Trial-by-trial changes in vel$_{max}$ averaged over participants for all groups. **d-e)** Trial-by-trial changes in vel$_{max}$ for **d)** Halo-R (orange) and Ctrl-R (red) and **e)** Halo-NR (light blue) and Ctrl-NR (dark blue) during *Training* on *Day1*. Regions shaded in grey indicate significant differences in performance between groups. **f-g)**

Trial-by-trial changes in **f)** FI and **g)** MT averaged over participants for all groups. **h)** Trial-by-trial changes in MT for Halo (pink) and Control (black) participants during *Training* on *Day1*. Shaded regions/error bars represent SEM.

to a global decrease in motor vigour, which was most pronounced in the Halo-R group. Converging results come from the LMM conducted separately for *Day2*, where we did not find a significant three-way interaction between *Reward*, *Trial Number*, and *Drug* ($t_{(84)}$ = 1.12, p = 0.266, S2 Table in S1 Appendix). Instead, we found significant main effects for both *Reward* ($t_{(84)}$ = 4.59, p < 0.001) and *Trial Number* ($t_{(84)}$ = 3.54, p < 0.001) on *Day2*.

3) Haloperidol does not affect the facilitatory effects of reward on movement fusion on *Day1*

Baseline analysis did not reveal any differences between groups in FI scores (ANOVA, Group: F = 2.66, p = 0.4478). Results from the LMM revealed a significant main effect for both *Reward* ($t_{(84)}$ = 3.27, p = 0.002, S3 Table in S1 Appendix, Fig 2F) and *Trial Number* ($t_{(84)}$ = 5.88, p < 0.001) on *Day1*. Importantly, we also found a interaction between *Drug*, *Reward* and *WM* ($t_{(84)}$ = 5.88, p < 0.001) on *Day1*, suggesting that haloperidol may differently modulate the reward-based effects on FI in participants with high and low WM. An ANOVA with *Reward*, *WM* and *Drug* as between factors revealed a significant main effect for *Reward* (F = 6.42, p = 0.0022) but no other main effects or interactions were found. Furthermore, LMM results revealed a significant interaction between *Drug*, *Trial Number* and *WM* ($t_{(84)}$ = -2.02, p = 0.049). Estimating marginal effects, we found that the participants with high WM that received haloperidol did not improve in FI over the course of *Training* (β = 0.09, CI [-0.1 0.2], p = 0.240). In contrast all other groups did (Halo low WM: β = 0.24, CI [0.1 0.4], p = 0.001; Ctrl high WM: β = 0.35, CI [0.2 0.5], p < 0.001; Ctrl low WM: β = 0.20, CI [0.04 0.35], p = 0.012). Post-hoc analysis including pairwise z-test comparisons revealed no significant differences between slopes. The results suggest that haloperidol does not affect the facilitatory effect of reward on movement fusion. Yet, it may have a subtle effect on movement fusion in participants with high WM. LMM results for Day2 revealed a significant main effect for both *Reward* ($t_{(84)}$ = 2.30, p = 0.006) and *Trial Number* ($t_{(84)}$ = 3.68, p < 0.001), but no other significant results were found (S4 Table in S1 Appendix).

4) Limited effect of haloperidol on movement times during Day1.

Using a Kruskal Wallis test, with group as a factor, we did not find any significant differences at Baseline (F = 5.44, p = 0.1424). Results from the LMM revealed main effects for *Reward* ($t_{(84)}$ = 45.78, p < 0.001, S5 Table in S1 Appendix, Fig 2G) and *Trial Number* ($t_{(84)}$ = 31.52, p < 0.001) on *Day1*. Importantly, we also found a significant interaction between *Trial Number* and *Drug* ($t_{(84)}$ = 4.22, p = 0.043). Estimating marginal effects, we found that the *Ctrl* groups exhibited a significant decrease in MTs over the course of Training (β = -0.37, CI [-0.5–0.2], p < 0.001). The Halo group also showed a decrease in MTs, but to a lesser extent (β = 0.17, CI [-0.3–0.04], p = 0.012). Post-hoc analysis revealed that the slopes were significantly different (z = 2.04, p = 0.041) suggesting that haloperidol led to a global slowing in the Halo groups. This aligns with the global slowing seen in vel$_{max}$. Further post-hoc analysis using a row-shuffle non-parametric permutation test, found that the Halo groups were significantly faster during early training (Fig 2H). Yet, this effect was limited to Trial 6–11, suggesting that over the course of *Training* they progressively slowed down. A separate LMM conducted for *Day2* did not reveal a significant *Trial Number* x *Drug* interaction ($t_{(84)}$ = 0.16, p = 0.873, S6 Table in S1 Appendix). Instead, we again found a significant main effect for both *Reward* ($t_{(884)}$ = -5.16, p < 0.001) and *Trial Number* ($t_{(84)}$ = -4.39, p < 0.001).

5) Haloperidol does not affect performance across *Post Assessments*.

Assessing performance across post assessment, we conducted separate mixed-effect ANOVAs for each day (i.e., *Day1*, *Day2* and *Day7*) for each outcome variable ($vel_{max}$, FI and MT).

In line with the previous results, we only found haloperidol to have a subtle effect on $vel_{max}$. Specifically, a significant interaction between *Reward*, *Timepoint* and *Drug* on *Day1* (F = 5.75, p = 0.0186, $\eta^2$ = 0.008) was found. To further assess this result, we calculated the difference in average performance between *post-Rew* and *post-NoRew* for each subject and ran separate Wilcoxon tests for each pairing (N = 4) and used FDR to correct for multiple comparisons. This post-hoc analysis revealed a significant difference in $vel_{peak}$ change from *post-Rew* to *post-NoRew* between Halo-R and Ctrl-R (Z = 2.87, p = 0.0166, $\eta^2$ = 0.42). However, no further post-hoc test reached significance. Interestingly, on *Day2* we found a significant main effect for *Drug* (F = 3.99, p = 0.0491, $\eta^2$ = 0.045). Yet, all other mixed-effect ANOVAs did not reveal any evidence that haloperidol affects movement fusion or overall movement time on any day.

## Discussion

Previous work was able to demonstrate that motor sequence learning is driven by distinct behavioural invigoration processes [10]. Specifically, it was shown that a monetary incentive led to an increase in motor vigour through increases in $vel_{max}$. In contrast, performance-based feedback enhanced movement fusion indexed as increases in fusion (FI). In the present study, we sought to investigate whether these behaviourally distinct reward-based processes rely on the same or dissociable dopaminergic mechanisms. An independent control experiment performed prior to the main experiment suggests that haloperidol was active during the sequential reaching task (positive control). Yet, our results are inconsistent with previous literature. Specifically, we did not find any strong evidence that haloperidol modulates the faciliatory effects of reward on movement fusion. This result diverges from previous research showing that a D2 antagonist impairs motor sequence learning [36]. Furthermore, we found that haloperidol led to a selective impairment in motor vigour, which led to a global slowing in movement times. This result aligns with previous work showing that a D2 antagonist impairs motor vigour during reward-effort decision-making [26, 44]. Therefore, our results suggest that a D2-antagonist differentially influences reward-based effects on movement vigour and movement fusion, indicating that the dopaminergic mechanisms underlying these two processes may be distinct.

### Haloperidol has a selective effect on the reward-based effects on motor vigour

Our results showed that the D2 antagonist haloperidol led to a selective impairment on motor vigour. Specifically, we found that $vel_{max}$ decreased over the course of Training on Day1, whereas $vel_{max}$ significantly increased in Ctrl-R which is in line with previous work on the reward-based effects on motor vigour [10]. Crucially, we did not observe any performance differences between the *NoRew* group. Thus, our results provide evidence that a D2 blockage impairs motor vigour which aligns with previous research [26]. Specifically, previous work investigating the role of DA during reward-effort integration showed that a D2 antagonist reduces the willingness to produce effortful squeezes in a isometric grip force task [24, 26, 28]. This is further in line with previous research showing that motor vigour can be enhanced with L-DOPA operationalising motor vigour as response speed [23]. Therefore, previous and our results suggest that motor vigour relies on a dopaminergic mechanism that can be modulated with a D2 manipulation.

### Haloperidol has a limited effect on movement fusion

In the present task, MTs could be reduced via two independent strategies: 1) an increase in $vel_{max}$ through and increase in motor vigour and 2) an increase in fusion measured as an increase

in FI. By reducing dwell times of reaching movement transitions, movement fusion allows for a faster but also smoother execution making the overall movement more efficient [10, 50]. Therefore, movement fusion represents a hallmark of motor sequence learning and has been shown to be often impaired in clinical populations such as stroke and PD patients [1–3]. DA has been implicated to underlie movement fusion and chunking with evidence coming from rodent [32–34], monkey [1, 35] and human [1, 3, 36–41] work. However, our results suggest that a D2-antagonist does not impair movement fusion. Specifically, we found that only participants with high WM that received haloperidol did not improve significantly in FI over the course of Training on *Day1*. However, post-hoc analysis showed that the rate of improvement in this group was not significantly different from the other groups. Additionally, our results do not show that haloperidol affects the reward-based effects on movement fusion. Post-hoc analyses aimed to further investigate the significant interaction between Drug*Reward*WM which only revealed a main effect for *Reward*. Therefore, our results did not find that haloperidol had a global effect on movement fusion, nor that it impairs the facilitatory effects of reward. This diverges from previous work that found impaired motor sequence learning during a button-press task using the D2-antagonist tiapride [36]. Previous work showed that DA may be implicated in the parsing of actions sequences [32], which may be more relevant in discrete button-press tasks than in continuous reaching tasks where movement fusion enhances reaching performance. In fact, most studies investigating the role of DA during motor sequence learning looked at motor sequence acquisition, which may be different to movement optimisation processes such as movement fusion. Nevertheless, our results highlight again that movement fusion is highly reward-sensitive sensitive, which suggests that there may be another dopaminergic mechanism underling it that does not heavily rely on D2 DA availability which may further be modulated by WM.

The most consistent finding with regards to dopamine underlying motor sequence learning comes from PD patients OFF medication [1, 37–39, 41]. In contrast to PD ON L-DOPA, PD patients OFF medication showed impaired motor sequence learning [1, 37–39, 41]. L-DOPA is the precursor to the neurotransmitter dopamine (as well as noradrenaline and adrenaline). It enters the central nervous system, where it is converted into dopamine. Therefore, compared to a D2-antagonist like haloperidol or a D2-agonist like cabergoline, L-DOPA increases dopamine levels whereas D2-antagonists/agonists modulate DA receptor activity [51]. Thus, there is accumulating evidence that motor sequence learning is highly reward-sensitive [10] and appears to rely on DA availability [1, 37–39, 41], yet to date it is still unclear which DA circuitries support this process in humans. Within this context, research employing neurotransmitter manipulations other than a DA blockage have also not produced consistent results [52]. Therefore, more research is warranted into the neural underpinnings of reward-based motor sequence learning.

In summary, in contrast to previous human work, we did not find strong evidence that a D2-antagonist modulates the reward-based effects on movement fusion during the production of sequential reaching movements. However, we found corroborating evidence that a D2-antagonist negates the reward-based effects on motor vigour. Taken together, our results suggest that a D2-antagonist differentially influences reward-based effects on movement vigour and movement fusion, indicating that the dopaminergic mechanisms underlying these two processes may be distinct.

## Supporting information

**S1 Checklist. STROBE statement—Checklist of items that should be included in reports of *cross-sectional studies*.**
(PDF)

**S1 Appendix. LMMs were conducted in JASP and included four fixed effects: 1. *Reward* (1 = Reward; 2 = No reward), 2. *Drug* (1 = haloperidol; 2 = placebo/control), 3. *Working Memory (WM;* 1 = high; 2 = low*)* and 4. *Trial Number* (1:180).** In the analysis, low WM (for the WM variable), No Reward (for the Reward variable), and haloperidol (for the Drug variable) were coded as the reference categories (i.e., 0). Therefore, looking at **S1 Table in S1 Appendix**, the results show that the Reward groups (*Rew*) produced higher maximum velocity (vel$_{max}$) compared to the No Reward (*NoRew*) groups. Additionally, the significant interaction between *Trial Number* and *Drug* indicates that the haloperidol groups (*Halo*) exhibited a decrease in vel$_{max}$ over the course of training compared to the *Ctrl* groups (since placebo/control is coded as 0).
(DOCX)

# Author Contributions

**Conceptualization:** Sebastian Sporn, Joseph M. Galea.

**Data curation:** Sebastian Sporn.

**Formal analysis:** Sebastian Sporn.

**Funding acquisition:** Joseph M. Galea.

**Investigation:** Sebastian Sporn.

**Methodology:** Sebastian Sporn.

**Project administration:** Sebastian Sporn, Joseph M. Galea.

**Resources:** Joseph M. Galea.

**Supervision:** Joseph M. Galea.

**Visualization:** Sebastian Sporn.

**Writing – original draft:** Sebastian Sporn.

**Writing – review & editing:** Sebastian Sporn, Joseph M. Galea.

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
