## [Decision Letter · Decision Letter 0]

23 Jun 2024

PONE-D-24-13233The effects of haloperidol on motor vigour and movement fusion during sequential reachingPLOS ONE

Dear Dr. Sporn,

Thank you for submitting your manuscript to PLOS ONE. After careful consideration, we feel that it has merit but does not fully meet PLOS ONE’s publication criteria as it currently stands. Therefore, we invite you to submit a revised version of the manuscript that addresses the points raised during the review process.

We look forward to receiving your revised manuscript.

Kind regards,

Claudia Brogna

Academic Editor

PLOS ONE

Journal Requirements:

4. Thank you for stating the following financial disclosure: "European Research Council"

5. Thank you for stating the following in the Acknowledgments Section of your manuscript: "This work was supported by the European Research Council starting grant: MotMotLearn (637488).

Please remove any funding-related text from the manuscript and let us know how you would like to update your Funding Statement. Currently, your Funding Statement reads as follows: "European Research Council".

Reviewers' comments:

Reviewer's Responses to Questions

**Comments to the Author**

1. Is the manuscript technically sound, and do the data support the conclusions?

Reviewer #1: Yes

Reviewer #2: Yes

2. Has the statistical analysis been performed appropriately and rigorously? 

Reviewer #1: Yes

Reviewer #2: Yes

3. Have the authors made all data underlying the findings in their manuscript fully available?

Reviewer #1: Yes

Reviewer #2: Yes

4. Is the manuscript presented in an intelligible fashion and written in standard English?

Reviewer #1: Yes

Reviewer #2: Yes

5. Review Comments to the Author

Reviewer #1: This paper examines the effect of a dopamine antagonist, haloperidol, on the motor performance measures, specifically lookin at two measures that are typically observed as a result of motor learning, namely a reduction in movement time and an increase in movement fusion. Contrary to expectations, they did not find strong evidence that the dopamine antagonist led to significant changes in either measures.

Generally, the paper is clearly written and rigorous, and easy to follow. Despite the mostly null results, I believe it is an important addition to the literature. Some questions and suggestions are listed below.

Consider chaning haloperidol to "a dopamine antagonist" in the title

Line 33 - abstract -consider mentioning that this task has previously be used to show reward-induced differences in vigour and fusion

line 36-37 - reward and no reward → probably reward or no reward would be clearer. I think it would make more sense to say there are four groups (2*2 design)

line 71 - explain what a D2 antagonist is

line 91 - again I would suggest reward OR no reward rather than and

line 94 - better to write the number of males and females (rather than assuming all non-males are females)

line 95 - what are the 3 experiments? do you mean three sessions?

line 96 - the two sentences starting with "All participants were ..." are repetitive, only one is needed (better the 2nd one)

line 103 - did any of the participants not receive medical approval? it is not obvious what the MD was checking

line 104 - 7 were left-handed, were all the rest right-handed (or were some ambidexterous?)

line 107 - maybe you mean counterbalanced rather than pseudorandomly (to ensure similar group sizes?)

line 174 - "and hence the box remained block" → it is not obvious which one results from the other

line 193 - what instructions were given during the "learning" phase (maybe familiarization is a better term?)

Figure 1 e - in my experience it is rare for "complete" fusion to take place (at least in a single day), maybe this would be clearer if an overlapping velocity profile was shown

line 239 - VWM - define what this is

line 269 - did you indeed find that there was a separation into "low" and "high" WM, i.e. did the distribution have two peaks rather than a single peak? Why should be expect low and high to both consist of exactly half the participants?

line 285 - there seems to be some contradiction here - MT is defined as the time from exiting the start box, whereas reaction time is when the start position exceeds 2cm (also it is not clear - exceeds 2cm from what?)

line 289 - I assume you mean you calculate the tangential velocity (or directionless speed)?

line 290 - 291 - it is not clear how the segments are divided - within 2cm define the start or end? first or last time?

line 297 - 298 - the definition of fusion (stop period between 2 movements gradually disappearing) is not clear - at some point they no longer stop but the velocity trough slowly increases over time (rather than the stop period gradually disappearing)

line 307 - the explanation in the text of the fusion index differs somewhat from the equation ([Disp-formula pone.0316894.e002]), which corresponds to a single "trough" whereas the text refers to 7 troughs. To make them equivalent, the equation could be changed to include a sum of i from 1 to 7, and then change 1 to i, and 2 to i+1 (and something similar for v_min)

line 308 - data exclusion - did you consider removing the trials also after "problematic" trials? These are likely also affected by the "problematic" trial or the dual task

line 382 - do you mean your measures were non-normal (rather than nonparametric), and hence you used nonparametric methods?

line 400 - is this analysis (until the end of the paragraph) now on the high WM group? it is not clear

line 414 - "we found no differences" - p>0.05 does not mean there are no differences, to make this claim a different test is needed (e.g. Bayesian ANOVA or two one-sided t-tests)

line 453 - the line ended without closing the brackets

line 488 - in the first paragraph of the discussion it may be helpful to remind the reader what FI is

line 508 (or elsewhere in the discussion) - it should be considered here or elsewhere why max vel and MT are not mirror images of each other. They are fairly close to mirror (see Figure 2 c and g), but where they differ may lead to some insight on what is happening here. For example, rather than increasing overlap between "submovements", there may instead be a "new" motor primitive created (as in Sosnik et al., 2004), which may mean the the FI is not the ideal measure

line 517 - I'm not sure whether this dichotomy between MT and fusion is necessarily the case (see above comment)

line 523 - are fusion and chunking the same thing necessarily?

line 527 - remind the reader what tiapride does

line 548 - the paper would be improved by introduction of a limitations section (particularly given the many surprising null results)

Reviewer #2: The paper investigates the effect of haloperidol, a dopamine antagonist, on motor vigour and movement fusion during a sequential reaching task. It examines whether dopamine-related mechanisms influence both motor vigour (peak velocity) and movement fusion in humans. Human participants, divided into groups based on a reward and drug condition, performed a sequential reaching task. Effects of reward, drug and trial on three dependent variables were assessed with LMM.

The study found that haloperidol had a significant effect on impairing motor vigour without affecting overall movement times. It also had no significant effect on movement fusion.

The study is very interesting and timely, it has a strong experimental design and the research idea is innovative, examining the dissociation of dopamine effects on peak velocity and movement fusion during motor sequence performance.

The manuscript is well written overall, and of interest to a wide community. My main concerns relate to the clarity of the methodological and results explanation.

Major points

- Throughout the manuscript and in the Discussion, the authors refer to the subtle/small effect of Haloperidol on the modulation of motor vigour/peak velocity. But no effect sizes are estimated. From the significant LMM interaction effect Drug * Trial *Reward, it would be good to assess marginal effects of Drug and Reward on the slope (DV by trial). I understand the statement “small effect” comes from doing trial-wise permutation tests and observing the effects exclusively in trials 170-180, but assessing the LMM slopes in the marginal effects of the three way-interaction seems more suitable.

- Response Accuracy analysis: Why not conduct a factorial analysis of RA including WM group, drug condition, and trial (Go/No-Go)?

- Do the authors expect that reaction time is not modulated by haloperidol?

- The explanation of LMM is not entirely clear.

- LMM: Please state the formula used in your analysis earlier, before specifying the implicit individual factor interactions implied by the main LMM.

- When discussing the introduction of random effects, the text could be clearer. State that, given the design of your study, you expect that each subject contributes variability in the slope of changes in DV across trials. Here, changes across trials would reflect training effects (improvements in performance). Why not consider random effects of subjects on the intercept? Which model explains the data better? One including/excluding random effects of subjects on the slope of DV by trial?

- Please elaborate on this: “within LMM has been a heated topic (49).” Why is it a heated topic? Don’t LMMs already provide statistical effects on fixed effects, including main effects and interactions?

- Why is this necessary? “This allowed us to investigate during which trials groups were exhibiting significantly different performance.” If participants performed 180 trials, do the authors expect to find between-group differences in some trials but not in others? Shouldn’t the collective behaviour of the DV across trials be simply parameterised as the intercept and slope? Comparing slopes between groups seems more meaningful than the DV on individual trials across 1:180.

- The post-hoc analyses following significant interaction effects should ideally be more extensive to assess all factor combinations and help us understand the implications of the interaction effect. For instance, page 16, lines 446-449: Would the interaction effect be explained by contrasting high and low WM groups separately for Reward groups (separately for each Drug group)?

- As mentioned, it is not clear why the permutation test separately for each trial is conducted, as effects of DV over time could be parametrised by the DV by trial slope, contingent on the other factors. The LMM could reveal these changes over time. Assessment of marginal effects could help here.

- It would help interpret the LMM results if the estimates from the Supp tables were referenced to the coding of groups. For instance, which Drug / Reward groups are coded as 0, and which ones a 1? With that information, the authors could easily interpret e.g. the negative and significant Trial*Drug interaction for peak velocity (-0.702, p = 0.049). If Placebo is 0, then this interaction estimate indicates that Halo relative to Placebo, decreases the slope of the DV change over trials.

- Discussion: Which evidence do the authors offer for this statement? “Nevertheless, our results highlight again that movement fusion is highly reward-sensitive sensitive, which suggests that there may be another dopaminergic mechanism underlining it that does not heavily rely on D2 DA availability.”

- The summary statement should be revised as small effects of haloperidol on peak velocity/motor vigour and no effects on movement fusion is not the same as “we did not find strong evidence that a

D2-antagonist modulates motor vigour and/or movement fusion during the production of sequential

reaching movements.”

Minor points

- Figure 1 seems to have broken typography.

- Figure 1e is unclear; the caption could be clearer. It seems that maximum velocity peaks are represented in the top left panel (blue traces), not in the top right panel. I do not see an illustration of '2) Fuse consecutive movements leading to increases in minimum velocities'. Which of the three panels shows an increase in minimum velocity? Why does the top right panel show only one movement instead of two, as shown in the background black traces?

- Page 11, something is missing here: 'Importantly, similarly mixed-model ANOVA with Drug (haloperidol vs placebo) as between factors and Feedback Condition (positive vs negative) as a within factor and RA as the dependent variable.' It appears incomplete.

- Page 13, distributions that do not meet the criteria for parametric statistical testing should indeed be analysed with non-parametric tests, but this sentence is not precise: 'We used one-sample Kolmogorov–Smirnov tests to test our data for normality and found that all measures were non-parametric.' Do the authors mean that 'and found that the data distributions were not suitable for parametric statistical testing, therefore …'?

- Clarify what you mean by 'Trial Number' and 'Trial'. Are these separate variables?

- Line 397: better phrased as 'while it impaired…'?

- Page 14, line 419: Where are the p-values from the permutation tests? Why do the authors argue that the effect was small/subtle? Did they estimate effect sizes?.

6. PLOS authors have the option to publish the peer review history of their article (what does this mean?). If published, this will include your full peer review and any attached files.

Reviewer #1: No

Reviewer #2: **Yes: **Maria Herrojo Ruiz

---

## [Author Response · Author response to Decision Letter 0]

16 Oct 2024

We have attached a 'Response to Reviewers' document

---

## [Decision Letter · Decision Letter 1]

18 Dec 2024

The effects of haloperidol on motor vigour and movement fusion during sequential reaching

PONE-D-24-13233R1

Dear Dr. Sebastian Sporn

We’re pleased to inform you that your manuscript has been judged scientifically suitable for publication and will be formally accepted for publication once it meets all outstanding technical requirements.

Kind regards,

Claudia Brogna

Academic Editor

PLOS ONE

Reviewers' comments:

Reviewer's Responses to Questions

**Comments to the Author**

1. If the authors have adequately addressed your comments raised in a previous round of review and you feel that this manuscript is now acceptable for publication, you may indicate that here to bypass the “Comments to the Author” section, enter your conflict of interest statement in the “Confidential to Editor” section, and submit your "Accept" recommendation.

Reviewer #1: All comments have been addressed

Reviewer #2: All comments have been addressed

2. Is the manuscript technically sound, and do the data support the conclusions?

Reviewer #1: Yes

Reviewer #2: Yes

3. Has the statistical analysis been performed appropriately and rigorously? 

Reviewer #1: Yes

Reviewer #2: Yes

4. Have the authors made all data underlying the findings in their manuscript fully available?

Reviewer #1: Yes

Reviewer #2: Yes

5. Is the manuscript presented in an intelligible fashion and written in standard English?

Reviewer #1: Yes

Reviewer #2: Yes

6. Review Comments to the Author

Reviewer #1: I am satisfied with the corrections made by the authors, and am happy to recommend the current version for publication.

Reviewer #2: The authors have done a great work addressing the comments. Their LMM analyses are robust and suitable to address their questions in this complex design. Overall, the manuscript has improved considerably. The study is imporant and will be relevant to the scientific community.

7. PLOS authors have the option to publish the peer review history of their article (what does this mean?). If published, this will include your full peer review and any attached files.

Reviewer #1: No

Reviewer #2: **Yes: **Maria Herrojo Ruiz

---

## [Editor Report · Acceptance letter]

23 Jan 2025

PONE-D-24-13233R1 

PLOS ONE

Dear Dr. Sporn, 

I'm pleased to inform you that your manuscript has been deemed suitable for publication in PLOS ONE. Congratulations! Your manuscript is now being handed over to our production team.

Kind regards, 

on behalf of

Dr. Claudia Brogna 

Academic Editor

PLOS ONE